# Low-Noise, Low-Power Readout IC for Two-Electrode ECG Recording Using Common-Mode Charge Pump for Robust 20-V$_{PP}$ Common-Mode Interference

**Kyeongsik Nam, Gyuri Choi, Mookyoung Yoo, Sanggyun Kang, Byeongkwan Jin, Hyeoktae Son, Kyounghwan Kim and Hyoungho Ko ***

Department of Electronics Engineering, Chungnam National University, Daejeon 34134, Republic of Korea
* Correspondence: hhko@cnu.ac.kr

**Abstract:** A low-noise and -power readout integrated circuit (IC) for two-electrode electrocardiogram (ECG) recording is developed in this study using a common-mode charge pump (CMCP) for a robust 20-V$_{PP}$ common-mode interference (CMI). Two-electrode ECG recording offers more comfort than three-electrode ECG recording. Contrasting to the three-electrode ECG recording, the two-electrode ECG recording is affected by CMI during measurements; the intervention of a large CMI will distort the ECG signal measurement. To achieve robustness for the CMI, the proposed ECG readout IC adopts CMCP—it uses switched capacitors that store and subtract CMI by control logic. In this paper, a window comparator structure is applied to CMCP to obtain a signal with less distortion. The window voltage ranges were set between the input common-mode ranges in which IA can operate. Therefore, a signal with less distortion was obtained by stopping the operation of CMCP between the window voltage ranges. It also reduced additional current consumption. To achieve this, the proposed circuit is implemented using a chopper stabilization technique. The chopper implemented in the amplifier can reduce low-frequency noise components, such as 1/f noise, and it comprises a CMCP, current feedback instrumentation amplifier, QRS peak detector, relaxation oscillator, voltage reference, timing generator, and serial peripheral interface on a single chip. The proposed circuit was designed using a standard 0.18 μm CMOS process with an active area of 0.54 mm². The proposed CMCP achieves a CMI robustness of 20 V$_{PP}$ at 60 Hz. The measured input-referred noise level was 119 nV/$\sqrt{\text{Hz}}$ at 1 Hz, and the power consumption was 23.83 μW with a 1.8 V power supply.

**Keywords:** low-noise; low-power; common-mode charge pump (CMCP); common-mode interference (CMI); window comparator; current feedback instrumentation amplifier (CFIA)

## 1. Introduction

Continuous monitoring of electrocardiograms (ECG) is used to diagnose cardiovascular diseases, such as cardiac arrhythmias, bradycardia, and paroxysmal supraventricular tachycardia. These diseases cause dizziness, shortness of breath, fatigue, and death [1–3]. Therefore, users require that the ECG monitoring devices be comfortable and portable for rapid disease detection and convenience. Thus, two-electrode monitoring is more popular than three-electrode monitoring [4–7].

When the instrumentation amplifier (IA) and human body are connected, power-line interference and noise causes the large common-mode interference (CMI) through the bio-impedance. The generated large CMI causes distortion and saturation of the ECG signal. Therefore, it is important to control large CMI. The ECG recording method to reduce CMI is as follows.

As illustrated in Figure 1a, the three-electrode ECG monitoring uses a two-electrode connection for ECG signal acquisition and a third electrode to bias the body by connecting it to the driven right leg (DRL) to prevent 60-Hz CMI [8,9].

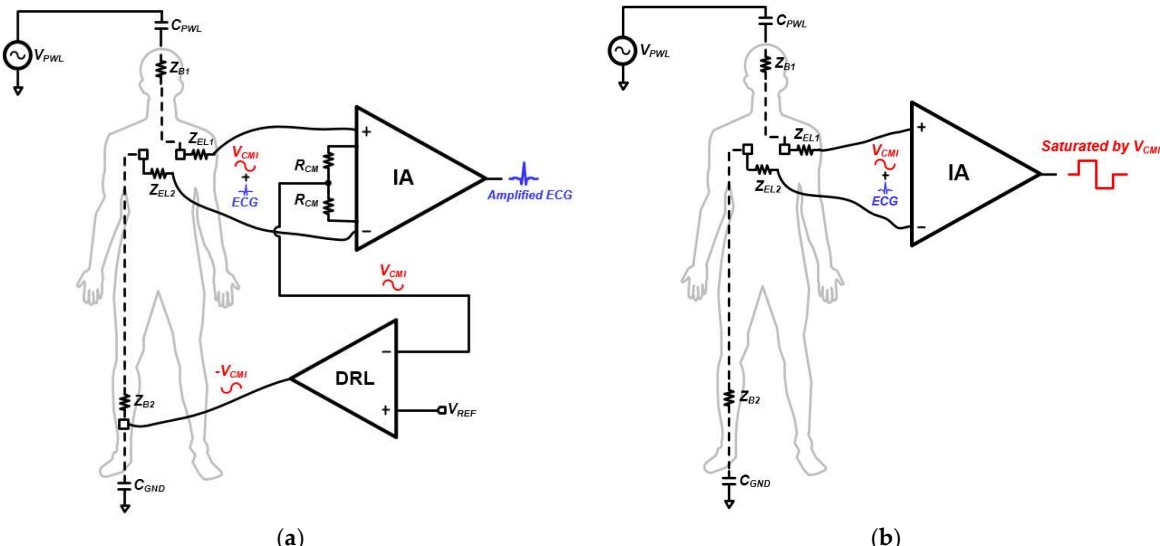

**Figure 1.** (**a**) Conventional three-electrode ECG monitoring; (**b**) two-electrode ECG monitoring.

The two-electrode ECG monitoring is shown in Figure 1b. This monitoring system, while eliminating the third electrode, provides users with better comfort. Although the area was increased in the CMCP added to reduce the CMI, the DRL scheme mainly used in three-electrode recording was not applied. Therefore, the overall area consumption did not increase significantly compared to previous studies. In addition, the number of electrodes is reduced compared to the three-electrode method. Compared to the three-electrode method, the area was reduced in the entire system.

However, the ECG signal is saturated by the CMI owing to the removal of the third electrode. This is because the proposed current feedback instrumentation amplifier (CFIA) cannot handle large common-mode input voltages because of the limited operating point of the input transistor. When the input exceeds the power supply range, as shown in Figure 2, the electrostatic discharge (ESD) diodes are turned on, and the signals are saturated. The common-mode charge pump (CMCP) schemes with single comparator to suppress the CMI were reported [4,5]. In these schemes, the switching noise and inter-modulation distortions caused by the CMCP operation can distort the ECG signals.

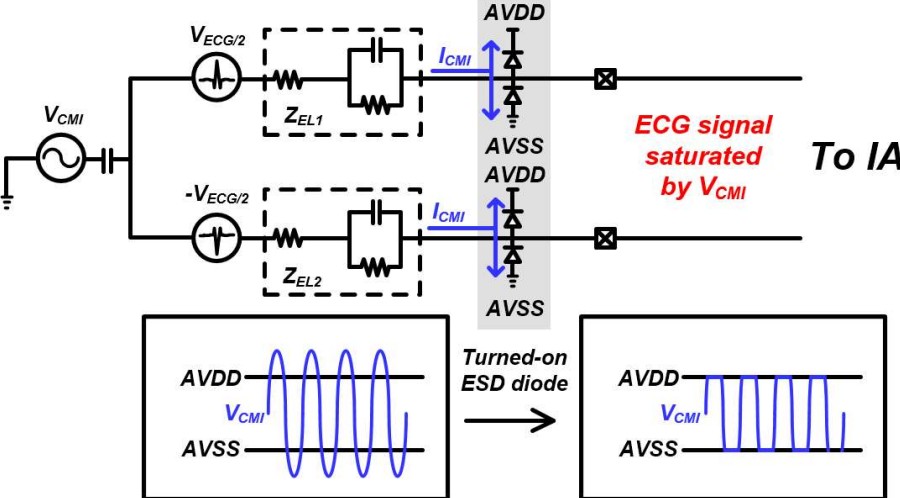

**Figure 2.** Two-electrode recording saturated by CMI without CMCP topology.

In this paper, the common-mode charge pump (CMCP) with window comparator is proposed. The CMCP can prevent turn-on of the ESD diodes and maintains a constant common-mode range that achieves the CFIA operating point.

The charge pump is enabled when the input common mode exceeds the desired common-mode window, which is monitored by the window comparator. If the input common mode is in the desired window, the input signals are not degraded by the charge pump operation. With the CMCP, the ECG input signals can be properly measured without saturation under large CMI conditions.

In addition, low-noise characteristics and the high-input impedance for high-accuracy characteristics are very important for amplifying a small ECG signal (1–2 mV) [10,11]. Therefore, we used the CFIA structure to achieve high-input impedance and adopted chopper stabilization in the CMCP loop and CFIA to reduce low-frequency 1/f noise.

Recently, wearable platforms including watches, armbands, and disposable patches have shown the capability to capture ECG signals in a convenient and effective way [12]. The ECG signal is one of the most important biophysical signals and can be used for the monitoring of daily activities, emotional states, and cardiovascular diseases. In recent research, the diagnosis systems and various applications based on the accurate classifications of ECG signals using the advanced artificial intelligence (AI) show superior performance [12]. The ECG waveforms contain much meaningful information; however, the most ECG acquisition circuits in the wearable platforms use the QRS peaks to count the heartbeat. So, we adopt QRS peak detector in the proposed circuit. The recently reported neuromorphic architectures for biophysical neural network can provide the advanced signal processing of ECG signals [13–18]. The AI-assisted ECG system can extend the current wearable and clinical ECG applications.

The proposed ECG readout integrated circuit (IC) for long-term monitoring is discussed as follows. Section 2 presents the top architecture and circuit implementation of the proposed CMCP, and a schematic of the CFIA. Section 3 presents the measurement results for the proposed ECG readout IC. Finally, Section 4 concludes the study with a performance summary and comparison.

## 2. Circuit Implementation

### 2.1. Top Architecture

The block diagram of the proposed low-noise and low-power readout IC for two-electrode ECG recording is shown in Figure 3. The proposed IC comprises a CMCP stage, amplifying stage, and sub-blocks in a single chip.

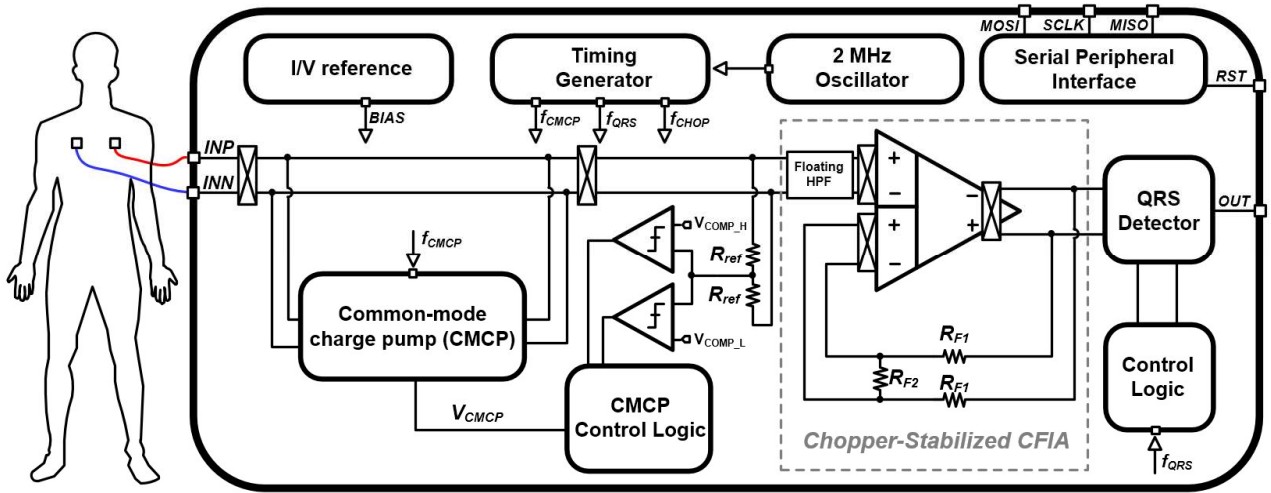

**Figure 3.** Block diagram of the proposed readout IC.

The CMCP stage comprises switched capacitors, CMCP control logic, and a window comparator. The amplification stage comprises a floating high-pass filter (HPF), chopper-stabilized CFIA, and QRS peak detector. The sub-blocks include the current/voltage reference, timing generator, 2 MHz oscillator, and serial peripheral interface (SPI).

In addition, since the CMRR characteristics are degraded because of electrode impedance mismatch, additional CMRR enhancement is required. Finally, additional measurements are needed by fabricating a small module for portable long-term recording.

In the CMCP stage, the input CMI is subtracted using switched capacitors, CMCP control logic, and a window comparator. The window comparator compares the hysteresis voltage range at $V_{COMP\_H}$ and $V_{COMP\_L}$ to maintain the common-mode range between $V_{COMP\_H}$ and $V_{COMP\_L}$.

CFIA amplifies the ECG signal with a high-input impedance characteristic between other IA structures [19–22]. CFIA uses a chopper stabilization technique to reduce low-frequency noise. The QRS peak detector detects the QRS peak of the ECG signal.

The internal operation parameters, including clock timing, register setting, CMCP control capacitor value, CMCP operation frequency, and chopper frequency, are programmable using the SPI.

### 2.2. Common-Mode Charge Pump Stage

The architecture of the proposed CMCP is illustrated in Figure 4. The CMCP comprises two switches, a CMCP capacitor ($C_{CMCP}$), buffer, and window comparator. The CMCP control logic creates an upward- or downward-shifting voltage to the common-mode input to maintain an acceptable input common-mode range of the CFIA.

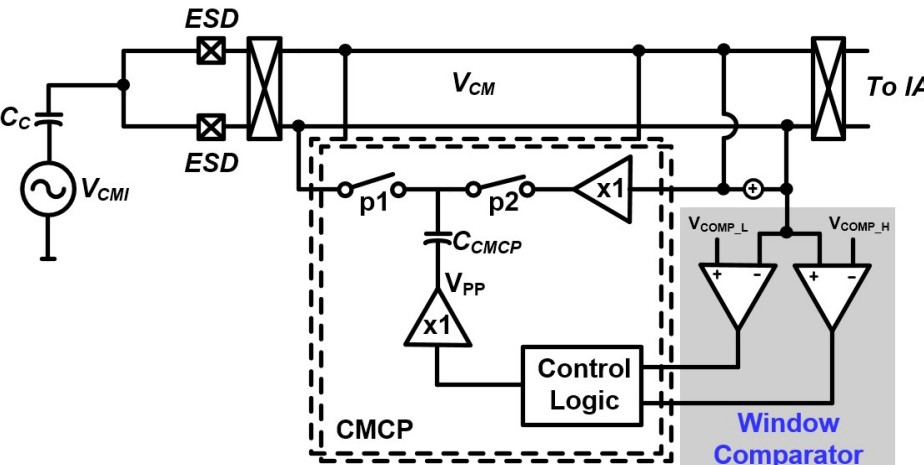

**Figure 4.** Top architecture of the proposed CMCP.

In this study, the CMCP operation was prevented within the input common-mode range by applying a window comparator, thereby reducing additional power consumption. The comparison voltage $V_{COMP\_H}$ of the window comparator was set to 1.1 V, the maximum operating point voltage of CFIA, and $V_{COMP\_L}$ was set to 0.7, the minimum operating point voltage.

Figure 5 shows the timing diagram of the CMCP operation. When $V_{CMI}$ increases more than $V_{COMP\_H}$, and if the amplitude of $V_{CMI}$ is higher than that of $V_{COMP\_H}$, the $V_{PP}$ is raised to a high (1.8 V) state. When p2 is high, the current state is maintained. Subsequently, when the p1 phase is converted to a high state, the common-mode voltage is maintained at $V_{COMP\_H}$ by downward-level shifting. When $V_{CMI}$ falls lower than $V_{COMP\_L}$, and if the amplitude of $V_{CMI}$ is lower than that of $V_{CMOP\_L}$, the $V_{PP}$ is reduced to a low (0 V) state. When p2 is in the high state, the current state is maintained. Subsequently, when the p1 phase is converted to a high state, the common-mode voltage is maintained at $V_{COMP\_L}$ by upward-level shifting.

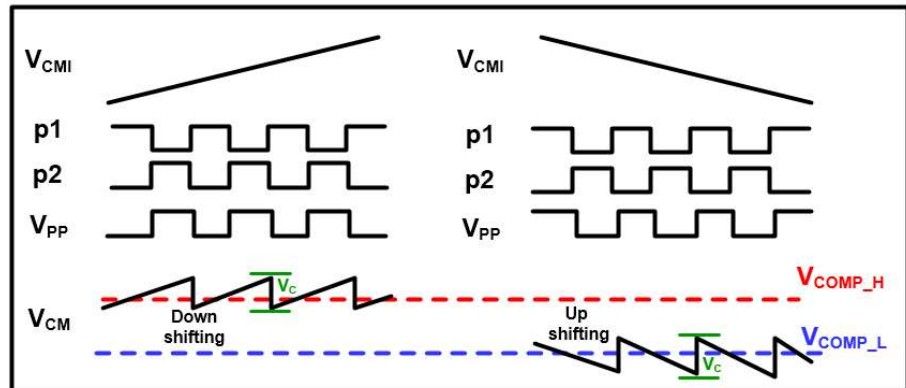

**Figure 5.** Timing diagram of the proposed CMCP.

The $V_C$ is level shift range is expressed as [4]:

$$V_C \approx V_{DD}\frac{2C_{CMCP}}{2C_{CMCP}+C_C} \tag{1}$$

where $C_{CMCP}$ is the CMCP feedback capacitor shown in Figure 4, and $C_C$ is between the $V_{CMI}$ and ESD pad. The proposed circuit operates at 1.8 $V_{DD}$, and the $C_{CMCP}$ value is set to 2 pF. The typical environmental parameter value [23] of $C_C$ is 220 pF, and the calculated $V_C$ was set to 32 mV. $C_{CMCP}$ was programmable to 2 pF and 10 pF. The $C_{CMCP}$ value was determined through a register setting and can be controlled by the SPI.

Figure 6 shows a schematic of the buffer. The buffer was applied driving the CMI voltage and $V_{PP}$ voltage. The buffer structure adopts a PMOS input stage and implements folded-cascode topology. Using folded-cascode topology can achieve a wide output voltage swing range. The total current consumption was 1.029 µA.

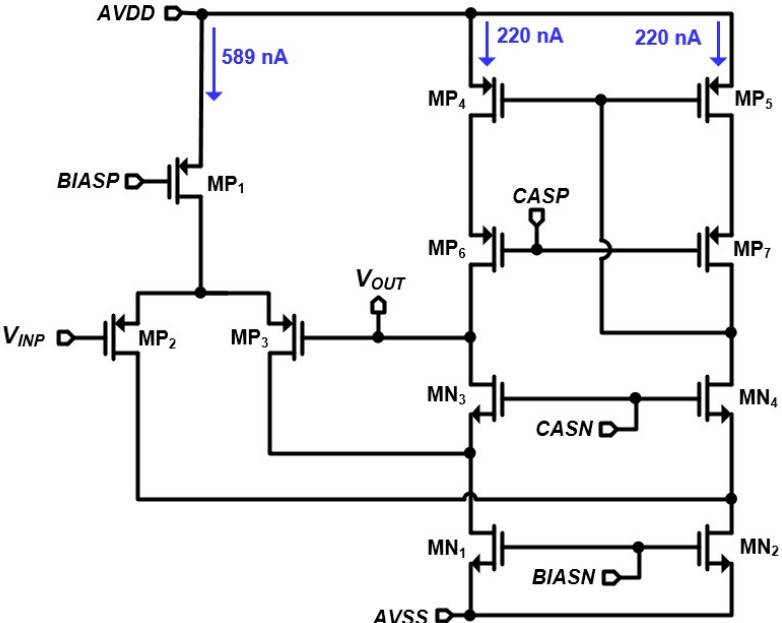

**Figure 6.** Schematic of the buffer.

The comparator structure is shown in Figure 7. The comparator comprised a two-stage preamplifier and latched comparator. The preamplifier implemented PMOS input single-stage cascade amplifier can reduce the kickback noise parameter. The latch implemented a CMOS latch to compare with the $V_{CMI}$.

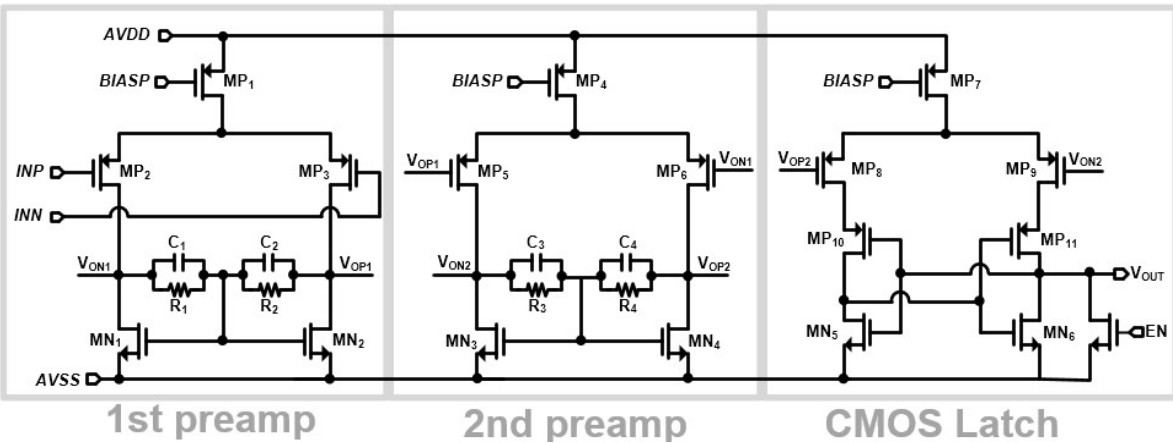

**Figure 7.** Schematic of the comparator.

### 2.3. Amplifying Stage

The amplification stage comprised a floating HPF, chopper-stabilized CFIA, and QRS peak detector. Figure 8 shows the structure of the floating high-pass filter. DC offset occurs during the chopper and CMCP operations. Therefore, a floating HPF was adopted to prevent a large DC offset. The high-pass filter is implemented using four pseudo resistors and two large capacitors ($C_{HPF}$) on the chip [24–26].

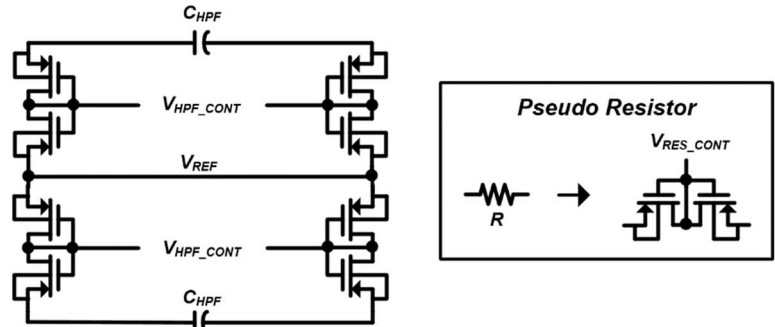

**Figure 8.** Structure of the floating HPF.

Figure 9 shows the block diagram of the chopper-stabilized CFIA. The CFIA comprised an input transconductance stage ($G_{m1-1}$, $G_{m1-2}$) and a Miller integrator ($G_{m2}$, $G_{m3}$). At the $G_{m2}$ and $G_{m3}$ stages, nested Miller compensation was implemented to achieve a fine frequency response [27,28].

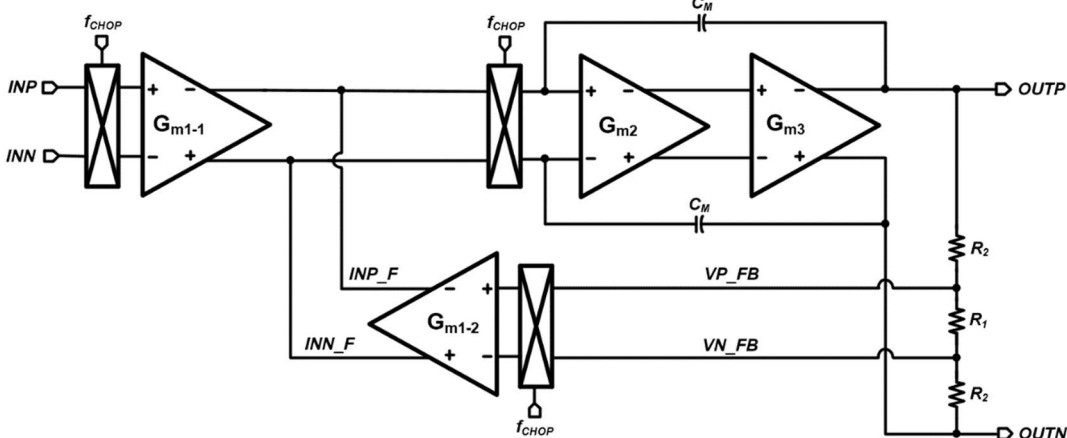

**Figure 9.** Block diagram of the proposed potentiostat readout circuit.

The transfer function of CFIA is expressed as:

$$\frac{V_{OUT}}{V_{IN}} = \frac{G_{m1\text{-}1}}{G_{m1\text{-}2}} \cdot \frac{2R_2 + R_1}{R_1} \tag{2}$$

where $G_{m1\text{-}1}$ is the input transconductance stage, $G_{m1\text{-}2}$ is the input feedback transconductance stage, and $R_2$ and $R_1$ are the feedback resistors. If $G_{m1\text{-}1}$ and $G_{m1\text{-}2}$ are well-matched and the ratio remains equal to one, the transfer function is given by:

$$\frac{V_{OUT}}{V_{IN}} \approx 1 + \frac{2R_2}{R_1} \tag{3}$$

In Figure 10, the proposed CFIA adopts the input and feedback stages of the PMOS differential pair. The intermediate stage was implemented using folded-cascode topology with Monticelli class-AB biasing, and the output stage was implemented in class-AB.

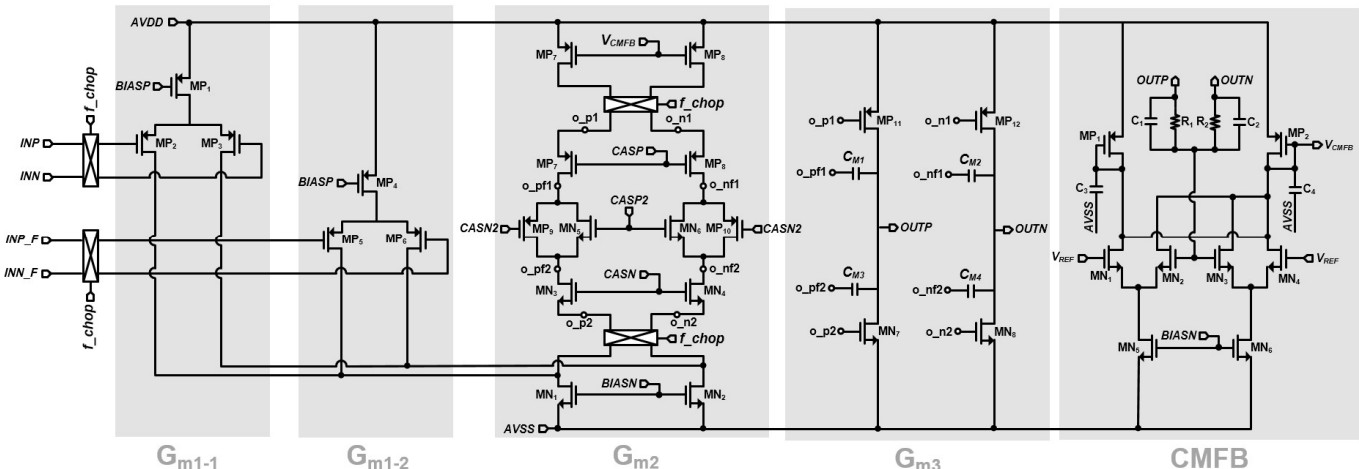

**Figure 10.** Schematic of the proposed potentiostat readout circuit.

In addition, a common-mode feedback (CMFB) circuit was implemented to stabilize the output common-mode voltage. Chopper stabilization was adopted to reduce the low-frequency 1/f noise. Choppers were implemented in front of the input stage and in front of the common-gate stage. The individual switch in the chopper was implemented using a CMOS transmission gate, and the chopping frequency of this circuit was 15 kHz.

The structure of the QRS peak detector is illustrated in Figure 11. The QRS peak detector comprises a comparator, AND gate, low-pass filter (LPF), and buffer [29]. A QRS peak was detected in the P-Q-R-S-T complex of the ECG signal. When comparing the ECG signal at INP and $V_{COMPOUT}$, when the R-term signal becomes higher than $V_{REF}$, $V_{COMPOUT}$ alternates from $V_{REF}$ (0.8 V) to the high state (1.8 V); when the comparator output voltage transitions from a high state (1.8 V) to a low state (0 V), the $V_{COMPOUT}$ signal changes back to VREF. Similarly, when comparing the ECG signal at INN and $V_{COMPOUT}$, when the R signal becomes lower than $V_{REF}$, $V_{COMPOUT}$ alternates from $V_{REF}$ (0.8 V) to the high state (1.8 V); when the comparator output voltage transitions from a low state (1.8 V) to a low state (0 V), the $V_{COMPOUT}$ signal changes back to $V_{REF}$. Finally, when the two comparator outputs passed through the AND gate, a QRS peak was detected.

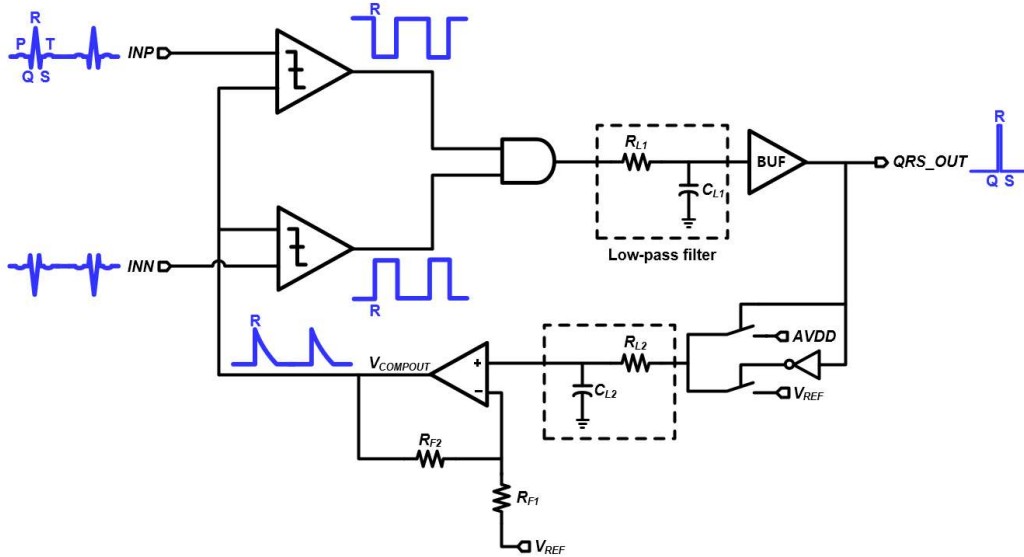

**Figure 11.** Structure of the QRS peak detector.

## 3. Measurement Results

A die photograph of the proposed readout IC for ECG recording is shown in Figure 12. The circuit was fabricated using a TSMC 180 nm CMOS process, and the active area of the proposed circuit was 856 × 1523 μm. The proposed circuit comprises a main block (CMCP, CFIA, and QRS peak detector), timing generator, bias block, and SPI in a single chip.

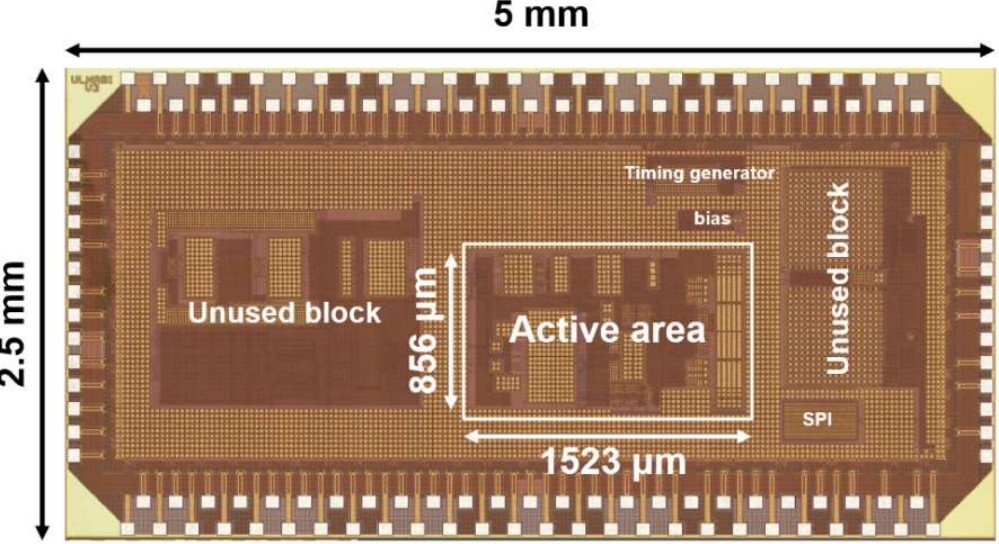

**Figure 12.** Die photograph of the proposed readout IC.

Figure 13 shows the test board with a chip-on-board (CoB) process and the measurement environment used to evaluate the performance of the proposed readout IC using a printed circuit board (PCB).

The gain error measurement results are shown in Figure 14. The CFIA gain was set as 46 dB (×200). When CMCP is enabled and disabled, the gain error is within 0.5% at a 1–50 Hz bandwidth.

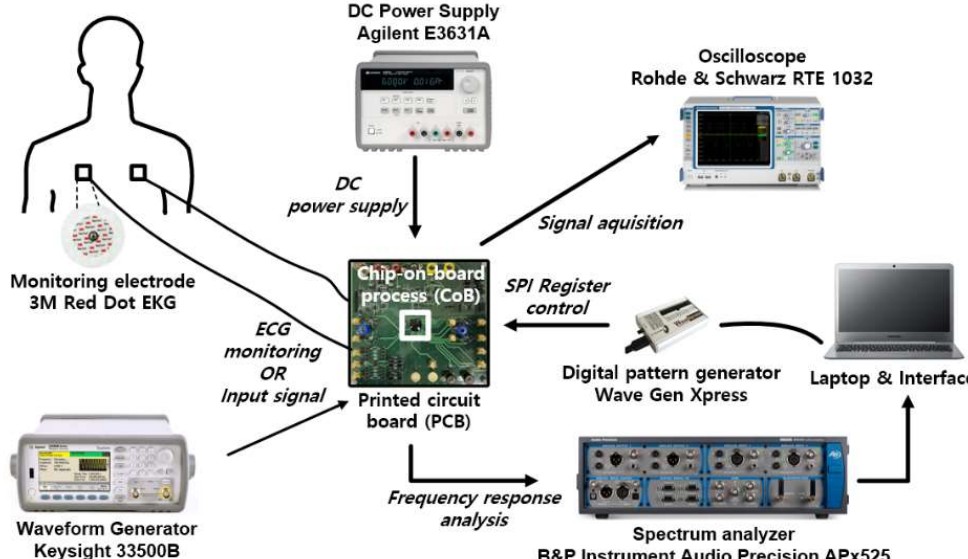

**Figure 13.** Measurement setup of the proposed ECG recording circuit.

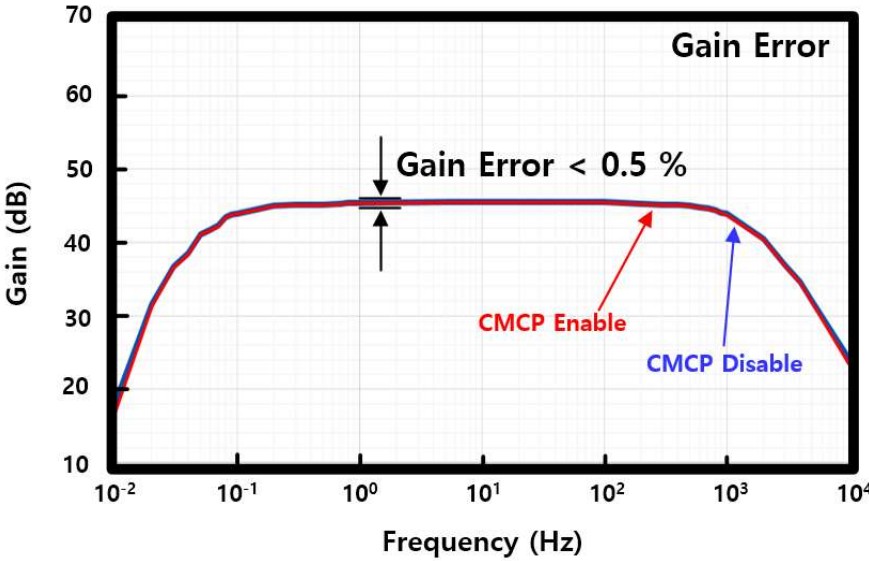

**Figure 14.** Gain error measurement results when the CMCP is enabled and disabled.

Figure 15 shows the measurement results of the CMCP operation. A 20-$V_{PP}$ at 60 Hz sinusoidal signal was applied as the CMI. The CMCP comparators were set to $V_{COMP\_H}$ = 1.1 V and $V_{COMP\_L}$ = 0.7 V. When the CMI increased, the CMCP created downward-level shifts so that the $V_{CMCP}$ maintained a 1.1 V. In addition, when the CMI was rising, the CMCP created downward-level shifts so that the $V_{CMCP}$ maintains a 1.1 V.

When the common-mode range was between 1.1–0.7 V, the CMCP operation was not performed, which has an advantage in terms of power consumption. The level-shifting voltage ($V_C$) was maintained between 30–34 mV. It can be seen that the value is similar to the calculated value (32 mV) in Section 2. The measured tolerance to the CMI result was a 20 V peak-to-peak $V_{CMI}$.

Figure 16 shows the measured input-referred noise of the proposed readout IC. The input-referred voltage noise was 119 nV/$\sqrt{Hz}$ at 1 Hz and 57 nV/$\sqrt{Hz}$ at 1 kHz. The measured integrated noise in the ECG signal band (1–100 Hz) was 0.59 $\mu V_{RMS}$.

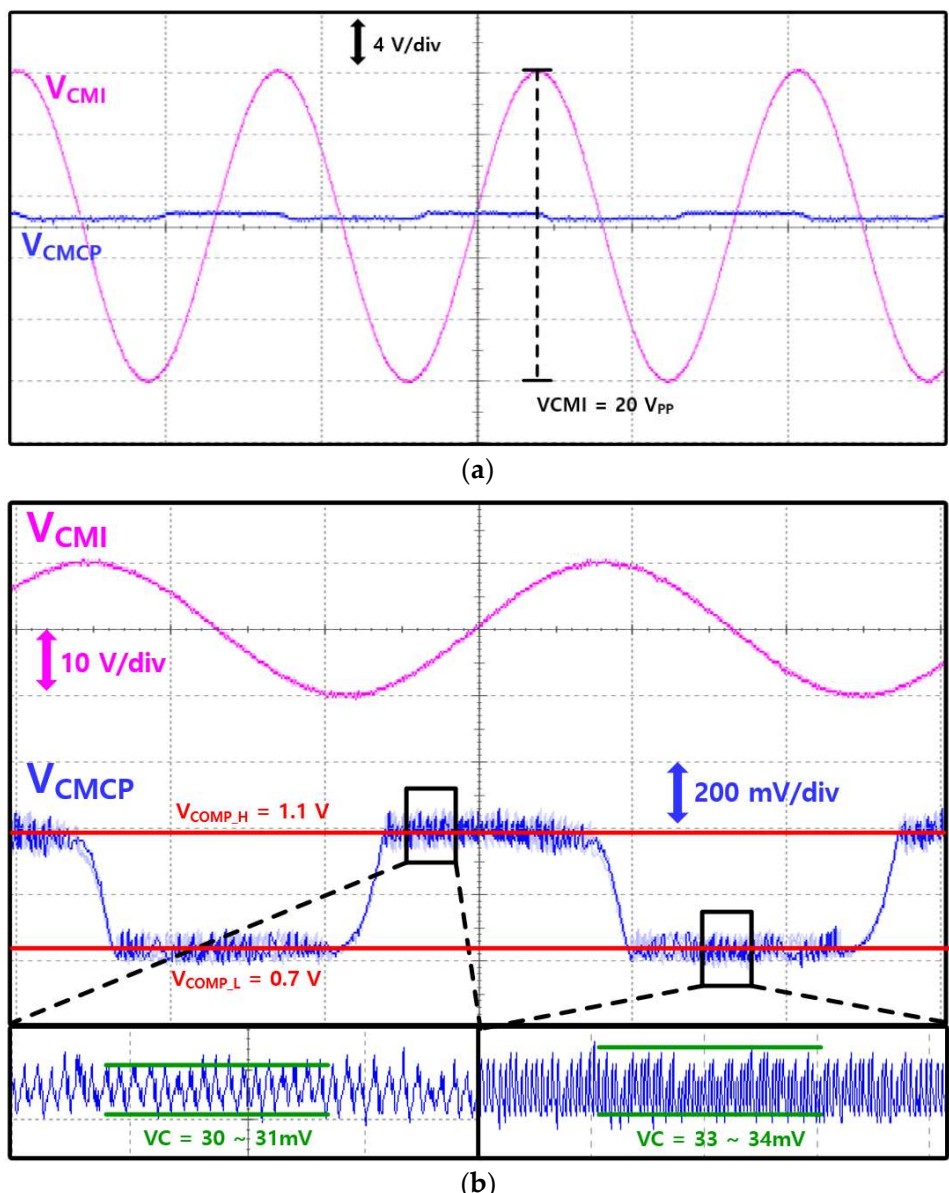

**Figure 15.** CMCP operation with 20-$V_{PP}$ CMI measurement results. (**a**) Compare with $V_{CMI}$ and $V_{CMCP}$ at same amplitude range; (**b**) magnified waveform.

Figure 17 shows the signal quality measurement results when the CMCP operation was enabled and disabled. A differential input of 2 mV, 10 Hz sinusoidal signal, and a CMI input of 1 $V_{PP}$, 60 Hz sinusoidal signal were applied. When the CMCP converts from the disabled state to the enabled state, the noise floor changed from −38 dB to −69 dB. The CMCP can have a good signal quality when the CMI voltage is presented.

The common-mode rejection ratio (CMRR) measurement results are shown in Figure 18. The average CMRR for 1–100 Hz was 51.4 dB. The CMRR is mainly determined by the impedance mismatch, and the CMRR of the CFIA has a small effect [4].

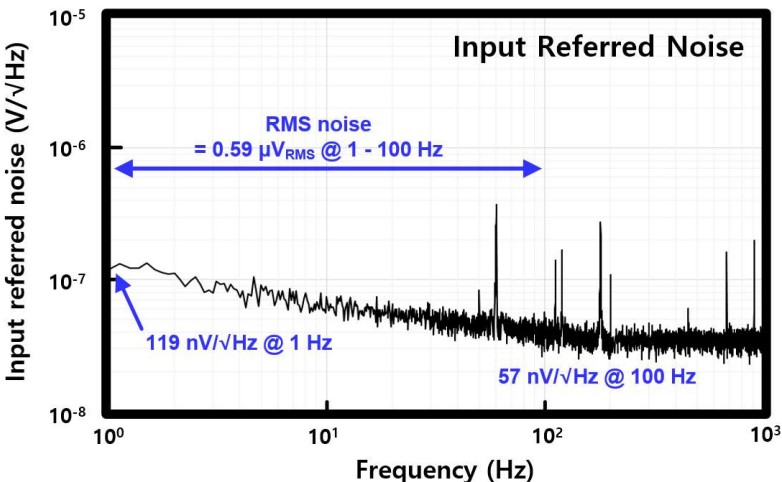

**Figure 16.** Input-referred noise measurement result of readout circuit.

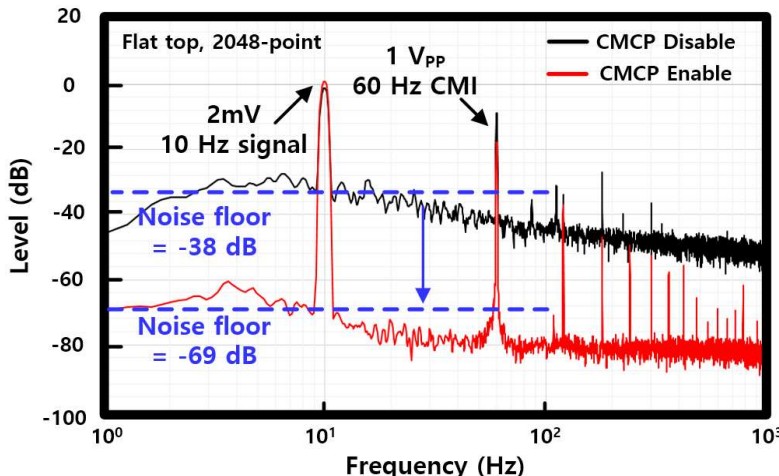

**Figure 17.** Signal quality measurement result when the CMCP operation is enabled and disabled.

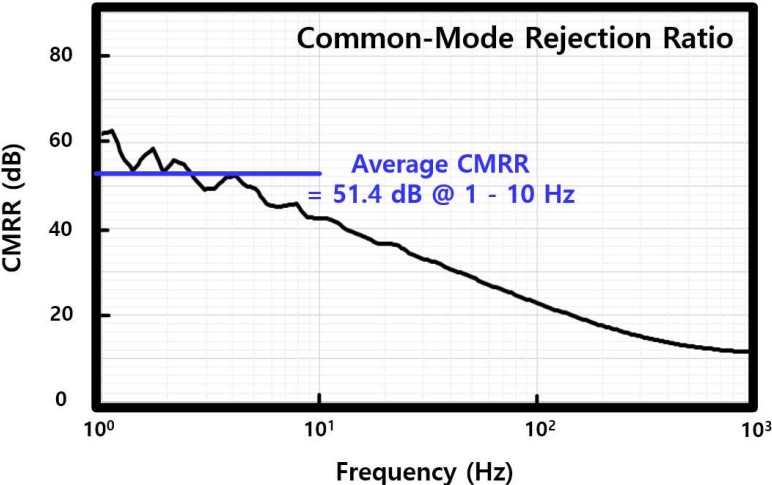

**Figure 18.** Common-mode rejection ratio measurement result of proposed readout circuit.

Figure 19 shows the ECG signal acquisition results. Figure 19a shows the waveform when the CMCP is disabled. When the CMCP is disabled, the ECG signal is saturated by the CMI, and the CFIA output does not exhibit any P-Q-R-S-T complex. Figure 19b shows

the waveform when the CMCP is enabled. An LPF with a cut-off frequency of 40-Hz was applied to the CFIA output. When the CMCP is enabled, the CMI is reduced by the CMCP operation and the CFIA output clearly shows the P-Q-R-S-T complex.

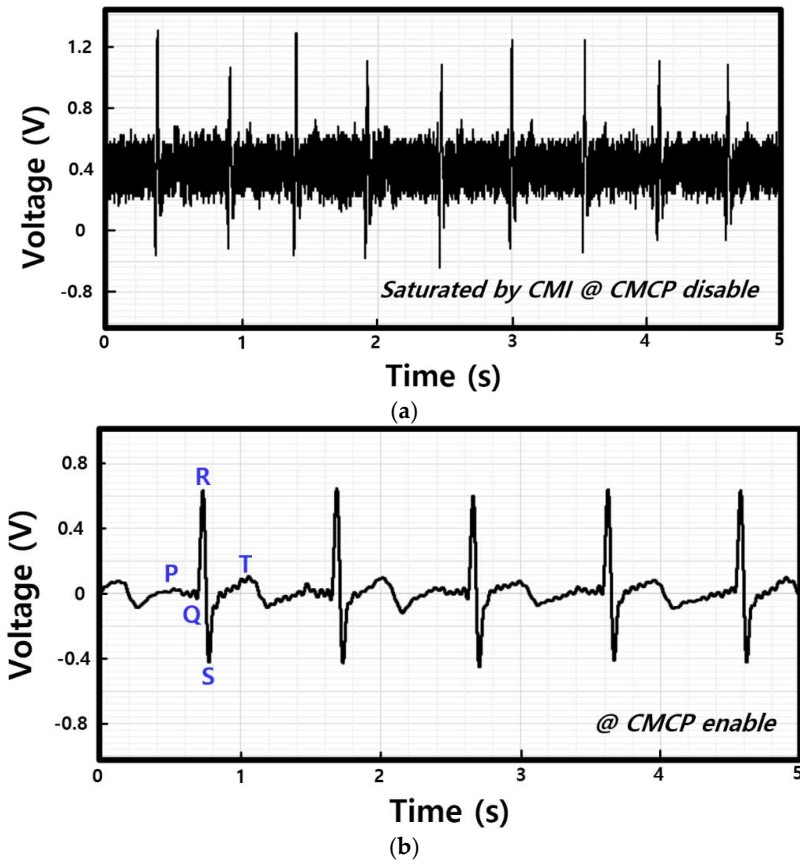

**Figure 19.** ECG signal acquisition result. (**a**) CMCP disabled; (**b**) CMCP enabled.

Table 1 summarizes the performance of the proposed readout IC for ECG recording using the CMCP [4,5,8,11,30,31]. The proposed readout circuit has low power consumption and low noise characteristics. In addition, the 20-$V_{PP}$ CMI tolerance characteristics can be verified.

**Table 1.** Performance summary between the proposed readout circuit and previous studies.

|  | IEEE JSSC 2021 [4] | IEEE ISSCC 2021 [5] | Appl.Sci 2020 [8] | IEEE ICMSP 2021 [11] | IEEE JSSC 2019 [30] | IEEE NEWCAS 2022 [31] | This Work |
|---|---|---|---|---|---|---|---|
| Process (μm) | 0.18 | 0.18 | 0.18 | 0.18 | 0.18 | 0.028 | 0.18 |
| Supply voltage (V) | 1.2 | 1.2 | 1.8 | 1.8 | 5 | 1.8 | 1.8 |
| Current consumption (μA) | 20.6 | 23.1 | 10 | 1.7 | 6.2 | 2.5 | 13.2 |
| CMI reduction structure | Single comparator CMCP | Single comparator CMCP | Feedback current generator | - | Active shield | - | Window comparator CMCP |
| Tolerance to CMI (V) | 15 | 30 | - | GND–VDD (respectively) | 2.8 | GND–VDD (respectively) | 20 |
| Input referred noise (μV$_{RMS}$) | 1.67 (1–100 Hz) | 5.05 (1–100 Hz) | 2.68 (1–108 Hz) | 2.96 (0.1–250 Hz) | 3.7 (0.5–100 Hz) | 10 (450–9.2 kHz) | 0.59 (1–100 Hz) |
| Bandwidth (kHz) | 10 | 15 | 0.107 | 0.250 | 20 | 8.75 | 2.7 |
| CMRR (dB) | 66 (Systematic) | 68 (Systematic) | 105 | N/A | 66 | 78 | 51.4 (1–10 Hz) |
| NEF [1] | 29.1 | 93.3 | 31.4 | 9.38 | 8.43 | 6.34 | 8.27 |

[1] noise efficiency factor (NEF) = $V_{ni}\sqrt{(2 \cdot I_{tot}/\pi \cdot U_T \cdot 4\ kT \cdot BW)}$.

## 4. Conclusions

This paper proposes a low-noise and low-power readout IC for two-electrode ECG recording using CMCP for robust CMI. In the two-electrode ECG measurements, it is important to reduce CMI. In this study, it was effectively reduced using a CMCP with a window comparator.

The proposed readout IC comprises a CMCP stage, amplifying stage, and sub-blocks in a single chip. The CMCP stage comprises switched capacitors, CMCP control logic, and a window comparator. The window comparator compares the input CMI with the reference voltage and transfers it to the CMCP control logic. Using the transmitted comparator output, a constant common-mode range is maintained through the control logic and switched capacitors. The amplification stage comprises a floating HPF, chopper-stabilized CFIA, and QRS peak detector. The floating HPF reduced the DC offset, the CFIA amplified the ECG signal, and the chopper-stabilization technique reduced the $1/f$ noise. The sub-blocks included the current/voltage reference, timing generator, 2 MHz oscillator, and SPI.

The proposed readout IC was fabricated by implementing a TSMC 0.18 μm CMOS process with an active area of $856 \times 1523$ μm. It achieved a CMI tolerance of 20 $V_{PP}$ at 60 Hz and consumed 23.83 μW of power at a 1.8 V supply voltage. The input-referred noise was 119 nV/$\sqrt{Hz}$ at 1 Hz and 57 nV/$\sqrt{Hz}$ at 1 kHz. The integrated noise in the ECG signal band (1–100 Hz) was 0.59 $\mu V_{RMS}$. The ECG signals were clearly obtained through measurements using the proposed circuit and achieved CMI tolerance of 20 $V_{PP}$ at 60 Hz. The proposed circuit with low-noise and CMI tolerance characteristics is suitable for two-electrode ECG recordings.

The main directions for future research are as below. This gain of the IA in this IC is fixed; however, in the real applications, including clinical or wearable platforms, the amplitudes of the ECG signals can show individual differences, and the high programmability for gain and bandwidth of the IA are required. In addition, since the CMRR characteristics are degraded because of electrode impedance mismatch, additional CMRR enhancement is required. Finally, additional measurements are needed by fabricating a small module for portable long-term recording.

**Author Contributions:** Conceptualization, K.N. and H.K.; methodology, H.K.; software, K.N.; validation, G.C., S.K. and B.J.; formal analysis, H.K.; investigation, M.Y. and H.S.; resources, H.K.; data curation, K.K.; writing—original draft preparation, K.N.; writing—review and editing, K.N.; visualization, K.N.; supervision, H.K.; project administration, H.K.; funding acquisition, H.K. All authors have read and agreed to the published version of the manuscript.

**Funding:** This work was supported by the Practical Technology development medical microrobot Program (R&D Center for Practical Medical Microrobot Platform, HI19C0642) funded by the Ministry of Health and Welfare (MOHW, Korea) and the Korea Health Industry Development Institute (KHIDI, Korea). This research was supported by the MSIT (Ministry of Science and ICT), Korea, under the ITRC (Information Technology Research Center) support program (IITP-2021-2017-0-01635) supervised by the IITP (Institute for Information & Communications Technology Planning & Evaluation). This work has supported by the National Research Foundation of Korea (NRF) grant funded by the Korea government (MSIT) (No.2022R1A2C100517011). This work is also supported by ABOV semiconductor, Inc.

**Institutional Review Board Statement:** Not applicable.

**Informed Consent Statement:** Not applicable.

**Data Availability Statement:** Not applicable.

**Acknowledgments:** The chip fabrication and EDA tool were supported by the IC Design Education Center (IDEC), Korea.

**Conflicts of Interest:** The authors declare no conflict of interest.

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
