# Peer review of "Low-Noise, Low-Power Readout IC for Two-Electrode ECG Recording Using Common-Mode Charge Pump for Robust 20-VPP Common-Mode Interference"

_applsci, doi:10.3390/app122412897_

Round 1
Reviewer 1 Report
The authors report a low-noise and low-power readout IC for two-electrode ECG 261 recording using CMCP for robust CMI. This looks a good system with a low-noise level and a low-power supply. However, the bandwidth of their system is not the best. So the authors may focus on improving the comprehensive performance of their system further. Here I ask the authors can consider more factors to improve, which are of significance for the application of their system. Also, please analyse the price cost, and the system they made is cheap enough compared to others? Please give more details in the main text for them I suggest here.
Author Response
Manuscript ID: applsci-2052592
December 02, 2022
MDPI Applied Sciences
Dear Editor
Enclosed please find the revised version of the manuscript entitled, " Low-Noise, Low-Power Readout IC for Two-Electrode ECG Recording Using Common Mode Charge Pump for Robust 20-VPP Common Mode Interference," being submitted for the MDPI Applied Sciences. I appreciate the valuable comments from the reviewers. I would also like to use the next page to answer the questions brought by the reviewers.
Thank you very much for your consideration.

Reviewer 2 Report
The author proposed a low noise and low power IC for ECG recording, the proposed IC is clear to understand and the comparison are provided with other work. It is an interesting topic with high quality, which can be accepted directly.
Author Response
Manuscript ID: applsci-2052592
December 02, 2022
MDPI Applied Sciences
Dear Editor
Enclosed please find the revised version of the manuscript entitled, " Low-Noise, Low-Power Readout IC for Two-Electrode ECG Recording Using Common Mode Charge Pump for Robust 20-VPP Common Mode Interference," being submitted for the MDPI Applied Sciences. I appreciate the valuable comments from the reviewers. I would also like to use the next page to answer the questions brought by the reviewers.
Thank you very much for your consideration.
Sincerely,
Hyoungho Ko
Ph.D.
enclosures
kn/HK

Reviewer 3 Report
This paper proposes a low-noise and low-power readout IC for two-electrode ECG recording using CMCP for robust CMI. In the two-electrode ECG measurements, it is important to reduce CMI. In this study, it was effectively reduced using a CMCP with window comparator. The paper is organized well. However, there are some points to be considered.
1. Abstract needs to be more precise highlighting major contributions.
2. It would be nice to explicitly list the future research directions. Please also discuss some limitations of your proposed method.
3. Some neuromorphic computing researches should be discussed, because neuromorphic computing is a novel computing paradigm with low power consumption and high-speed response, including: Scalable digital neuromorphic architecture for large-scale biophysically meaningful neural network with multi-compartment neurons; Neuromorphic context-dependent learning framework with fault-tolerant spike routing; BiCoSS: toward large-scale cognition brain with multigranular neuromorphic architecture; CerebelluMorphic: large-scale neuromorphic model and architecture for supervised motor learning.
4. The background of the proposed study should be further explained in detail. Some concepts are hard to comprehend without explaining clearly.
5. Grammar is expected to be further improved. Please check the manuscript carefully to remove the typos, improve the language and format.
Author Response

(The authors gave the same response as above.)

Reviewer 4 Report
The authors have performed an impressive study where they have proposed a low-noise and low-power readout integrated circuit for two-electrode electrocardiogram 261 recording using common-mode charge pump for robust common mode interference. The results are robust and will be useful to the community. Couple of points before final publication.
The introduction can be improved for a broader audience
The description of the schematic of the buffer can be improved
The explanation of Fig. 19 in connection with the results was not particularly clear to me.
Thank you
Author Response

(The authors gave the same response as above.)

Reviewer 5 Report
Reviewer’s suggestions:
1.A comfortable two-electrode electrocardiogram (ECG) with low-noise and -power 0.18-μm CMOS readout integrated circuit (IC) including a window comparator using a common-mode charge pump (CMCP) with chopper stabilization technique including CMCP, CFIA, and QRS peak detector), timing generator, bias block, and SPI was proposed in this research. However, the research performance of the proposed common-mode charge pump (CMCP) with window comparator including CMRR, input-referred noise, and measured integrated noise should be compared with the using of common mode charge pump (CMCP) schemes with single comparator and three-electrode electrocardiogram (ECG), which designed circuit structure of all referred references should be exhibited and compared in Table 1.
2.The section present of “2.2. Amplifying Stage” should be revised by the “2.3. Amplifying Stage”.
Author Response

(The authors gave the same response as above.)

Round 2
Reviewer 3 Report
I think it is important and meaningful to introduce some neuromorphic hardware implementation research as a compensate of the presented study. Some representative work can be introduced in the Introduction section, and it is critical to broaden the background of the paper and increase the potential research direction. These work includes: Scalable digital neuromorphic architecture for large-scale biophysically meaningful neural network with multi-compartment neurons; Neuromorphic context-dependent learning framework with fault-tolerant spike routing; BiCoSS: toward large-scale cognition brain with multigranular neuromorphic architecture; CerebelluMorphic: large-scale neuromorphic model and architecture for supervised motor learning.
Author Response
Manuscript ID: applsci-2052592
December 05, 2022
MDPI Applied Sciences
Dear Editor
Enclosed please find the revised version of the manuscript entitled, " Low-Noise, Low-Power Readout IC for Two-Electrode ECG Recording Using Common Mode Charge Pump for Robust 20-VPP Common Mode Interference," being submitted for the MDPI Applied Sciences. I appreciate the valuable comments from the reviewers. I would also like to use the next page to answer the questions brought by the reviewers.
Thank you very much for your consideration.
Sincerely,
Hyoungho Ko
Ph.D.
enclosures
kn/HK

Round 3
Reviewer 3 Report
I am satisfied with the revision. No comments remained.